# Vibronic mixing enables ultrafast energy flow in light-harvesting complex II

Eric A. Arsenault [1,2,3], Yusuke Yoneda [1,2,3], Masakazu Iwai [3,4], Krishna K. Niyogi [3,4,5] & Graham R. Fleming[1,2,3 ✉]

Since the discovery of quantum beats in the two-dimensional electronic spectra of photosynthetic pigment-protein complexes over a decade ago, the origin and mechanistic function of these beats in photosynthetic light-harvesting has been extensively debated. The current consensus is that these long-lived oscillatory features likely result from electronic-vibrational mixing, however, it remains uncertain if such mixing significantly influences energy transport. Here, we examine the interplay between the electronic and nuclear degrees of freedom (DoF) during the excitation energy transfer (EET) dynamics of light-harvesting complex II (LHCII) with two-dimensional electronic-vibrational spectroscopy. Particularly, we show the involvement of the nuclear DoF during EET through the participation of higher-lying vibronic chlorophyll states and assign observed oscillatory features to specific EET pathways, demonstrating a significant step in mapping evolution from energy to physical space. These frequencies correspond to known vibrational modes of chlorophyll, suggesting that electronic-vibrational mixing facilitates rapid EET over moderately size energy gaps.

[1] Department of Chemistry, University of California, Berkeley, CA 94720, USA. [2] Kavli Energy Nanoscience Institute at Berkeley, Berkeley, CA 94720, USA. [3] Molecular Biophysics and Integrated Bioimaging Division, Lawrence Berkeley National Laboratory, Berkeley, CA 94720, USA. [4] Department of Plant and Microbial Biology, University of California, Berkeley, CA 94720, USA. [5] Howard Hughes Medical Institute, University of California, Berkeley, CA 94720, USA. ✉email: grfleming@lbl.gov

Excitation energy transfer (EET) within light-harvesting complex II (LHCII) lies at the heart of green plant photosynthetic light-harvesting[1]. LHCII can comprise up to 50% by weight of the grana membrane, the site of water splitting by Photosystem II[2]. The complex is generally found as a trimer with each monomer containing eight chlorophyll (Chl) a and six Chl b molecules along with four carotenoid molecules[3]. The empirical Hamiltonian developed by Novoderezhkin and van Grondelle accounts for a wide range of linear spectroscopic measurements and forms the basis for theoretical and experimental studies of the EET dynamics and mechanism[4]. The spatial arrangement of the Chls and experimental linear absorption spectrum along with the exciton energy levels from the Novoderezhkin and van Grondelle Hamiltonian are shown in Fig. 1 (exciton energy levels are colored to match pigments in Fig. 1a that have the most substantial contribution).

From the earliest ultrafast spectroscopic measurements on LHCII, it was clear that rapid Chl b to Chl a transfer occurred[5,6] and as multidimensional spectroscopy has developed, models of EET within LHCII have advanced significantly[7–12]. At the same time, long-lived oscillatory features were observed, initially in the FMO complex[13] and later in LHCII[9]. Since then, the origin of these oscillations has been extensively debated[14–26]. Beyond this debate lies an even more challenging question—how does the observation of these beats spectroscopically connect to the mechanistic function of natural light-harvesting? For example, the assignment of beats as vibronic in origin does not necessarily demonstrate that electronic-vibrational mixing influences energy transport. In response to such a question, the mechanistic role of vibronic coupling in facilitating EET has also received significant theoretical attention, however, the conclusions can vary widely as there remains no uniform treatment of electronic-vibrational mixing[19–27]. To this end, recent work has expressed how certain aspects of vibronic mixing must be treated carefully, namely Yeh

et al.[22] illuminated the need to synchronously treat electronic energy fluctuations and vibrational relaxation in mixed electronic-vibrational systems, while Zhang et al.[19] highlighted the potential for a disconnect between spectral signatures and system dynamics by showing that both Condon and non-Condon coupling can lead to prolonged oscillatory features, but only non-Condon effects led to enhancements in electronic-vibrational energy transfer. The commonly employed technique of two-dimensional electronic spectroscopy (2DES) often lacks the spectral resolution to untangle the complex, congested spectra of relevant systems—making it challenging to discern whether or not an oscillatory feature is electronic, vibrational, or vibronic in nature[21]. Although recent work has successfully unveiled the presence of vibronic coupling in natural and artificial light-harvesting systems[28–34], the newly developed technique of two-dimensional electronic-vibrational (2DEV) spectroscopy[7,35–44], by focusing on vibrational transitions in the final light-matter interaction, has the potential to provide significantly improved experimental input into the interplay of electronic and nuclear dynamics in ultrafast energy (and charge) transfer. Several aspects of 2DEV spectroscopy are significant in this regard. The improved spectral resolution along the detection (infrared) axis enables features that are not discernable in degenerate (all electronic) spectra to be resolved. A further advantage of this extreme two-color technique is that in certain cases (e.g. this work), the spectral dynamics will be completely free of beats that are purely vibrational in origin, as will be discussed below (it should be noted that this is also possible to achieve with certain polarization schemes in 2DES[45]). This allows for the issue of the origin of oscillatory features to be easily distinguished, thus emphasis can be shifted to the actual role that these beats may serve. Finally, the center line slopes (CLSs) of the spectral features being related to a cross correlation of vibrational and electronic dipoles, as opposed to the autocorrelation relationship of conventional, degenerate

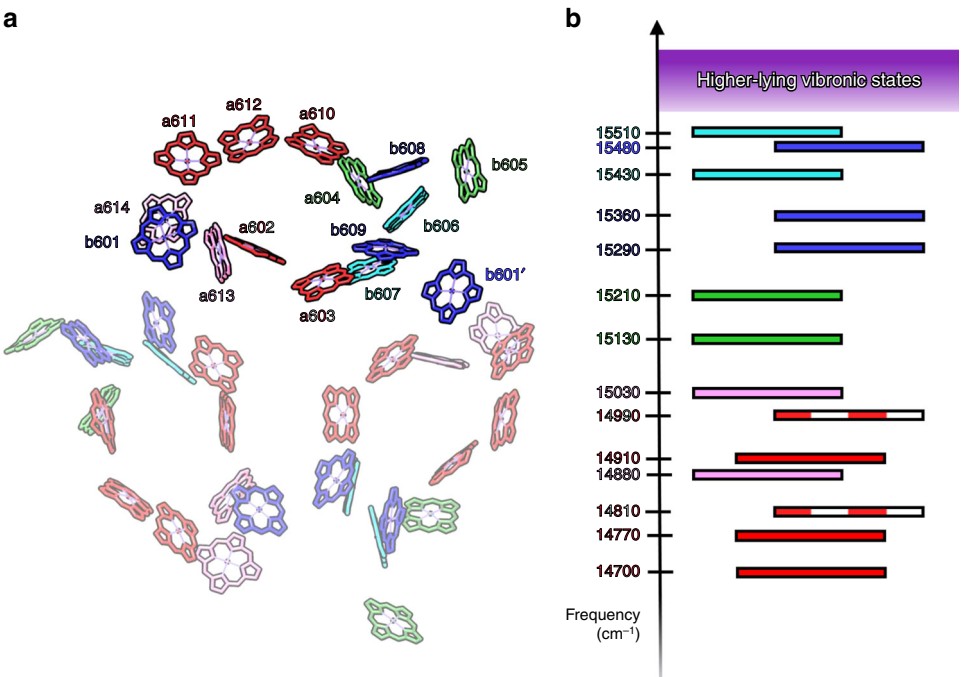

**Fig. 1 LHCII trimer structure and excitonic energy level diagram. a** Pared-down LHCII trimer from 2.72 Å X-ray crystal structure[3]. Chl a and Chl b pigments in one monomer are labeled (note: a prime indicates a pigment located in a different monomer). **b** Excitonic states of LHCII, which are colored to match the pigments with dominate contributions, based on the Novoderezhkin and van Grondelle Hamiltonian[4]. Excitonic states colored with pink, red and white, or red have predominately Chl a character, while those labeled with light blue or blue have predominantly Chl b character. The corresponding frequencies are based on past experimental work[9].

multidimensional spectroscopies, contains unique information on the mixing and dynamics of specific levels.

This work examines the sub-picosecond (<700 fs) energy transfer dynamics of LHCII at 77 K with 2DEV. Throughout, comparisons to a basic heterodimer model, featuring two electronically coupled monomers, each with one electronic degree of freedom (DoF) and one vibrational DoF, will be used to create an intuitive picture for much of the phenomena that control the observed spectral dynamics. Specifically, evidence for the direct participation of higher-lying vibronic states during the EET process will be discussed, in addition to the assignments of oscillatory signals, observed in both the peak intensities and CLS dynamics. The assignments provide a deeper understanding of the interplay between different sites in LHCII—directly connecting the intricate energetic, spatial, and vibronic landscapes of this pigment-protein complex. We will conclude by discussing the likely role of non-Condon effects in the EET dynamics of LHCII, which demonstrates the important next step towards determining the actual mechanistic function that quantum beats may serve during EET.

## Results and discussion

**Manifestation of EET in 2DEV spectra**. In a 2DEV experiment, visible pump pulses prepare an ensemble of electronic/vibronic states that evolve as a function of waiting time, $T$, and are then tracked with an infrared probe pulse. The resulting data are presented as (visible) excitation frequency-(infrared) detection frequency correlation plots at given waiting times. Another way to describe a 2DEV experiment is in terms of more traditional pump-probe spectroscopy—2DEV is the two-dimensional analog to a one-dimensional visible pump-infrared probe experiment, where the resulting 2DEV spectra will have spectral resolution along both the excitation and detection axes, as opposed to the visible pump-infrared probe experiment, which will only have spectral resolution along the detection axis. A more detailed description of the technique can be found in previous work[35].

In this work, we have chosen to probe modes in the higher frequency region (1525–1715 cm$^{-1}$), which do not explicitly promote energy transfer, but rather serve as spectators that report on the energy transfer dynamics. These modes were chosen specifically because in the case of Chl, these modes are highly local and thus not anharmonically coupled to the lower frequency modes (<1500 cm$^{-1}$)[46]. One major advantage of probing in this region, as opposed to the lower frequency region, which is energetically closer to the exciton energy gaps, is that since there is no anharmonic coupling between low and high frequency vibrational modes, the dynamics observed via probing the higher frequency modes are free from the modulation effects caused by the creation of low(er) frequency vibrational wavepackets. The practical translation of this is that any observed beats in the 2DEV spectra will only be electronic/vibronic in origin. A control 2DEV experiment performed on cresyl violet, which was chosen because it is well known to exhibit very strong vibrational coherences with well characterized frequencies in 2DES experiments[47,48], is provided in Supplementary Note 1. For the same reason explained above, no vibrational coherences are observed in the 2DEV spectra of cresyl violet when the higher frequency modes are probed.

Selected 2DEV spectra of LHCII are shown in Fig. 2. Along the excitation axis, the features centered around 14,800 cm$^{-1}$ and 15,500 cm$^{-1}$ correspond to the excitonic bands with mainly Chl $a$ character and mainly Chl $b$ character, respectively, while the two bands above 15,600 cm$^{-1}$ correspond to vibronic transitions that involve both Chl $a$ and $b$. Prominent in the spectra are two sets of excited state absorption (ESA) quartets labeled one through four and five through eight in Fig. 2b, where only the latter will be the focus of discussion (assignments of the other features not

discussed here can be found in ref. [7]). The same unique quartet structure is pronounced in the heterodimer model (Fig. 3b, c). The assignment of these quartets in the model is straightforward and translates easily to the experimental spectra of LHCII (model parameters chosen to be comparable to those expected for LHCII, see Supplementary Note 2 for more details). In the heterodimer model, electronic coupling between two monomers, each with one electronic DoF and one vibrational DoF, leads to the formation of six excitonic states (Fig. 3a). Excitonic states $|A\rangle$ and $|B\rangle$, which have mainly electronic character, are excited initially, resulting in the appearance of an A excitonic band and B excitonic band along the excitation axis. The other four excitonic states have mainly vibrational character and as such, $|C\rangle$ and $|D\rangle$ form the manifold of A excitonic states, whereas $|E\rangle$ and $|F\rangle$ form the manifold of B excitonic states. Thus, ESAs labeled C and D are transitions within the A excitonic band (between $|A\rangle \rightarrow |C\rangle$ and $|A\rangle \rightarrow |D\rangle$ in Fig. 3a, respectively), whereas ESAs denoted E and F are transitions within the B excitonic band (between $|B\rangle \rightarrow |E\rangle$ and $|B\rangle \rightarrow |F\rangle$ in Fig. 3a, respectively). The remaining ESAs—C′, D′, E′, and F′—are a manifestation of $|B\rangle$ to $|A\rangle$ or $|A\rangle$ to $|B\rangle$ EET and are cross peak-type features. For example, if $|B\rangle$ is initially excited, the B excitonic band will appear along the excitation axis, however, $|B\rangle$ to $|A\rangle$ EET during the waiting time will cause frequencies specific to the manifold of A excitonic states to appear in the B excitonic band along the detection axis because $|A\rangle \rightarrow |C\rangle$ or $|A\rangle \rightarrow |D\rangle$ will be the probed transitions, as opposed to $|B\rangle \rightarrow |E\rangle$ or $|B\rangle \rightarrow |F\rangle$. An increasing degree of mirror symmetry between bands along the excitation axis (Fig. 3b, c) in 2DEV indicates population transfer. Returning to LHCII, spectral assignments now become much easier. At $T = 50$ fs (Fig. 2a), the intensity of ESA five, located in the mainly Chl $a$ band at 1600 cm$^{-1}$, and seven, located in the mainly Chl $b$ band at 1590 cm$^{-1}$, are greatest, whereas ESA six and especially eight are barely distinguishable. At longer waiting times (Fig. 2b–d), the formation of four distinct ESAs at positions five through eight becomes increasingly evident as six and eight grow in. This indicates that ESAs five and seven have mainly Chl $a$ and $b$ character, respectively. Thus, ESAs six and eight are cross peaks indicative of EET between states of mainly Chl $a$ and mainly Chl $b$ character.

Shifting attention to higher excitation frequencies in the LHCII spectra, direct evidence for the participation of higher-lying vibronic states during EET appears in much the same way as for the $Q_y$ states. The detection frequencies in the higher-lying vibronic bands mirror those in the lower-lying Chl $a$ and $b$ bands, as excitation initially populating the higher-lying vibronic states is rapidly transferred to the lower-lying Chl $a$ and $b$ states. At longer waiting times (Fig. 2b–d), more population is funneled to the lower energy Chl $a$ and $b$ states, counterintuitively causing these higher-lying vibronic bands to grow in. One particularly revealing feature along the detection axis is the ground state bleach at 1690 cm$^{-1}$—highly specific to Chl $b$—present in the higher-lying vibronic bands[7]. This directly shows that these bands have mostly Chl $b$ character and indicates that EET directly from the higher-lying vibronic states → Chl $a$ states is not the predominant pathway of energy flow. Rather, transfer from the higher-lying vibronic states → lower-lying Chl $b$ states → Chl $a$ states contributes most to the vibronic energy funnel mechanism. Clearly, the nuclear DoF play an important role in LHCII, as the participation of the higher-lying vibronic states allows for an extension of efficient light-harvesting into a higher excitation frequency regime.

**CLS dynamics**. 2DEV offers another unique way to further elucidate both the complex energy transfer dynamics and the role of

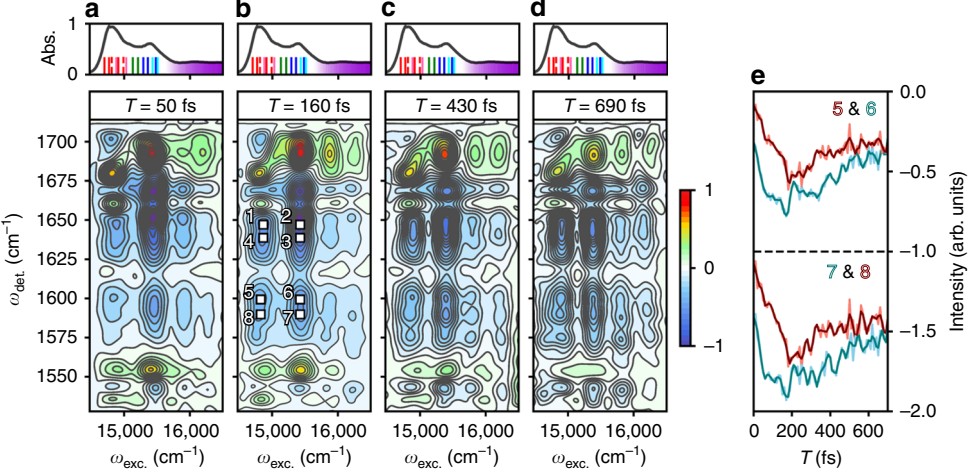

**Fig. 2 Two-dimensional electronic-vibrational spectra of LHCII at 77 K. a–d** 2DEV spectra of LHCII at waiting times of $T = 50$, 160, 430, and 690 fs, respectively. Positive features indicate ground state bleaches and negative features indicate excited state absorptions (ESAs). The intensities of all spectra have been normalized to $T = 0$ fs. Contour lines are drawn in 6.6% intervals. The region of the linear absorption spectrum of LHCII (at 77 K) that was excited during this experiment (the $Q_y$ bands), along with the excitonic states described in Fig. 1, has been placed at the top of each of the 2DEV spectra in (**a–d**). The grouping of red and pink lines distinguishes the excitonic band of mainly Chl $a$ character, while the grouping of blue lines distinguishes the excitonic band of mainly Chl $b$ character. The purple continuum highlights the region of the absorption spectrum composed of the higher-lying vibronic states of mixed Chl character. In (**b**), ESA features of interest have been labeled. **e** Example of the observed oscillatory intensities, for ESAs five through eight. The intensity dynamics for seven and eight have been offset for clarity. The unfiltered intensities are shown in light red and blue, while the data in dark red and blue has been subjected to a 1200 cm$^{-1}$ cutoff filter, in order to highlight the lower frequency oscillations of interest.

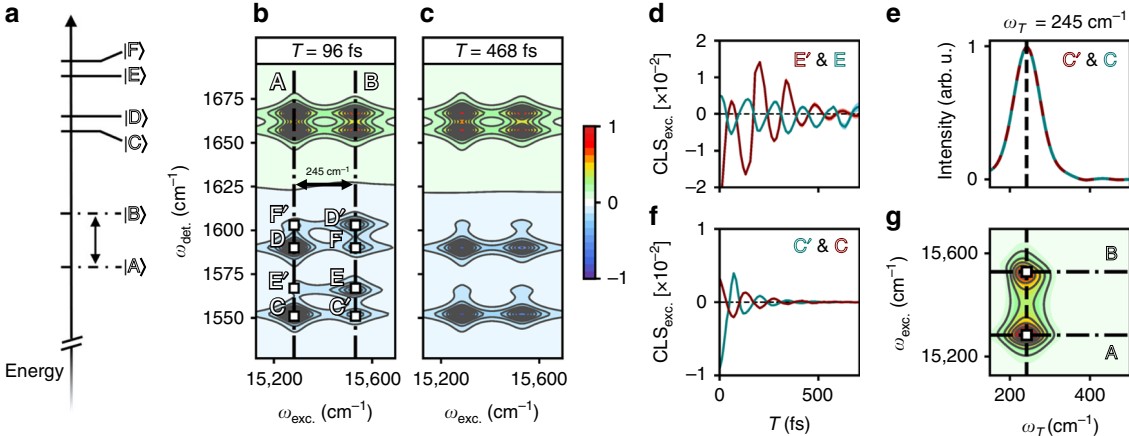

**Fig. 3 2DEV spectral evolution of a model heterodimer. a** Energy level diagram for the six excitonic states of the heterodimer model, where the ground state manifold has been omitted for clarity. **b, c** 2DEV spectra for the model at $T = 96$ fs and $T = 468$ fs. Positive features indicate ground state bleaches and negative features indicate excited state absorptions. All spectra have been normalized to $T = 0$ fs. Contour lines are drawn in 8.3% intervals. Features of interest have been labeled C, D, E, F, C′, D′, E′, and F′. The two bands along the excitation axis have been marked by dashed-dotted black lines and labeled by the excitonic state that they originate from (**A** and **B**). The $|A\rangle$ to $|B\rangle$ energy gap has also been labeled. **d, f** Center line slope dynamics along the excitation axis (CLS$_{exc.}$) of certain features, colored according to the peak labels in the top right corner of each plot. **e** Magnitude of the cross-power spectrum of the CLSs of features C′ and C, where the oscillatory frequency of 245 cm$^{-1}$ is marked by a dotted black line (a peak in the cross-power spectrum indicates a shared frequency). **g** Intensity beat map along the excitation axis of features C′ and C, where the oscillatory frequency of 245 cm$^{-1}$ is marked by a dotted black line and the involved excitation frequencies are labeled and marked by black dotted-dashed lines.

the nuclear DoF during the dynamics in the form of CLSs. In a 2DEV experiment, the CLS is a sensitive measure of the correlation between the nuclear and electronic DoF. Throughout, we will focus on CLSs along the excitation axis, CLS$_{exc.}$ (CLSs along the detection axis contain identical information). As described previously, the CLSs of asymmetric features are best determined based on conditional averages, $f(\omega_{exc.})$, given by ref. [36]:

$$f(\omega_{exc.}) = \frac{\int d\omega_{det.} P(\omega_{exc.}, \omega_{det.})\omega_{det.}}{\int d\omega_{det.} P(\omega_{exc.}, \omega_{det.})} \quad (1)$$

where $P(\omega_{exc.}, \omega_{det.})$ is the spectral area that encloses the peak and is calculated at each $\omega_{det.}$ in $P(\omega_{exc.}, \omega_{det.})$. The CLS$_{exc.}$ is then extracted from a linear fit of the conditional averages of the peak intensity versus $\omega_{exc.}$ ref. [36]. The CLS is a dynamical quantity and is calculated as a function of waiting time. To understand how energy transfer manifests in the CLS, we return to the model. Shown in Fig. 3d, f are the CLS$_{exc.}$ of the lower frequency ESA quartet. The oscillatory behavior of the CLSs is striking and results from $|B\rangle$ to $|A\rangle$ transfer, made unmistakable by the fact that the oscillation frequency is equal to the energy gap between

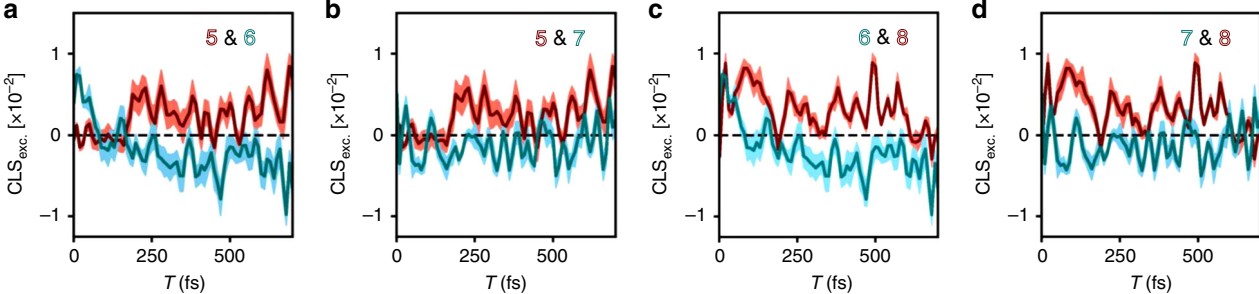

**Fig. 4 Center line slope dynamics of LHCII trimer.** Time domain center line slope (CLS) dynamics along the excitation axis of features five through eight. Pairings have been made to emphasize the striking complementary behavior. The pairs of features shown are **a** five and six, **b** five and seven, **c** six and eight, and **d** seven and eight. The CLSs are colored according to the peak labels in the top right corner of each plot. Shaded light red or blue regions around each CLS indicate the standard error from the linear fits used to calculate the CLS$_{exc.}$.

these two excitonic states. Using the model ESAs C and C′ as an example, the frequency relationships between this pair of CLSs can be more clearly evaluated by calculating the magnitude of the cross-power spectrum, given by the magnitude of the conjugate Fourier transform of one CLS multiplied by the Fourier transform of the other (Fig. 3e), where by taking the magnitude, only frequency relationships are retained. The cross-power spectrum of the C and C′ CLSs demonstrate that these features share a common frequency of 245 cm$^{-1}$ ($|B\rangle$ to $|A\rangle$ energy gap). It is worth noting here that the CLS has one major advantage in characterizing energy transfer because, as we have shown, it is free of the population dynamics that greatly complicate cross-peak amplitudes. Therefore, when applied to electronically coupled systems, the CLS is an electronic/vibronic coherence-specific measurement. Turning back to LHCII, Fig. 4 compares the CLS$_{exc.}$ dynamics of several spectral features. The CLS$_{exc.}$ dynamics of each feature clearly exhibit long-lived (>700 fs), strongly oscillatory correlations between the electronic and nuclear DoF—a signature of EET in 2DEV. LHCII, obviously a far more complicated system than the heterodimer model, has an abundance of EET pathways and energy gaps, which translates directly to the presence of many oscillatory frequencies in the CLS dynamics. By simultaneously taking advantage of (1) the sensitivity of the CLS to EET, (2) a spectrally resolved excitation axis (the E in 2DEV), (3) the high spectral resolution along the detection axis (the V in 2DEV), (4) the observation of EET-driven intensity beats in the ESA features, and (5) the fact that the probed modes will not be modulated by purely vibrational coherences, the origin of the oscillatory features can be unambiguously determined.

**Assignments and discussion of oscillatory frequencies.** We will now demonstrate how to extract information pertaining to the specific excitonic levels involved in the EET process. Starting with the model, by taking slices along the excitation axis at a detection frequency of 1555 cm$^{-1}$ (through ESAs C and C′), and then performing a Fourier transform along the waiting time, an intensity beat frequency map can be created (Fig. 3g). In the beat frequency map are two peaks, each with a beat frequency of 245 cm$^{-1}$, at the A and B band excitation frequencies. This same frequency appears in the cross-power spectrum of the corresponding C and C′ CLSs (Fig. 3f) and in both cases it appears as a result of EET between $|B\rangle$ and $|A\rangle$. However, if we did not already know the origin of the oscillatory frequency, by observing it in the cross-power spectrum between C and C′, we would learn that it originates from EET. To facilitate an assignment of the specific states involved, we would next look for the excitation frequencies in the intensity beat map of C and C′ that have an energy gap equal to that oscillatory frequency. In 2DEV spectra, those

excitation frequencies are the origin of the oscillatory feature. This procedure can be applied to the experimental 2DEV spectra of LHCII (particularly using ESAs seven and eight), in order to assign the origin of oscillatory features (Fig. 5). In our analysis below, we have chosen to focus on the 475–700 cm$^{-1}$ region because towards the lower beat frequency region, the concurrent loss of frequency resolution and large number of closely spaced excitonic states makes explicit assignment of these beats difficult. Therefore, the oscillatory frequencies that will be discussed are 475, 570, 650, and 700 cm$^{-1}$ (marked by vertical dashed lines in Fig. 5), which have been assigned to EET between excitonic states of mainly Chl $a$ character and states of mainly Chl $b$ character. To make more site-specific assignments of the origins of these beats, the energy levels from Fig. 1b (color coded to match) were overlaid on the intensity beat map. For example, the beat frequency of 650 cm$^{-1}$ (Fig. 5c, g) is shared by excitation frequencies of ~15.360 cm$^{-1}$ (b601′-b608-b609) and ~14,700 cm$^{-1}$ (a610-a611-a612), which have an energy gap of 660 cm$^{-1}$. This indicates that the beat frequency of 650 cm$^{-1}$, found in both the peak intensity and CLS dynamics, originates from b601′-b608-b609 → a610-a611-a612 transfer. However, unlike the model, also evident in the beat frequency map is the participation of excitation frequencies in addition to those that have an energy gap matching the beat frequency. This is an indication that other states are involved in that particular energy transfer process. For example, there is evidence for the participation of higher-lying vibronic states, which have already been established as energetically connected to the lower-lying Chl states. The beat frequency of 650 cm$^{-1}$ appears at the higher vibronic level excitation frequencies as a result of these states rapidly populating the lower-lying b601′-b608-b609 state that then undergoes b601′-b608-b609 → a610-a611-a612 transfer. Additionally, the presence of other excitation frequencies that nearly match those in Fig. 1b result from the complexity of the energetic landscape of LHCII—in addition to transfer between excitonic states of Chl $b$ character to those of mainly Chl $a$ character, intradimer, intratrimer, dimer to trimer, and dimer to dimer Chl $a$ to Chl $a$ or Chl $b$ to Chl $b$ energy transfer can occur[4,49]. Through the same analysis, the beat frequency of 700 cm$^{-1}$ (Fig. 5d, h), common to excitation frequencies of ~14,810 cm$^{-1}$ (a602-a603) and ~15,510 cm$^{-1}$ (b606-b607), is assigned to b606-b607 → a602-a603 transfer. Lower frequency beats, 475 cm$^{-1}$ (Fig. 5a, e) and 570 cm$^{-1}$ (Fig. 5b, f), are more difficult to assign due to the participation of many excitation frequencies, but 475 cm$^{-1}$ is a compelling candidate for either b606-b607 → a613-a614 or b601′-b608-b609 → a602-a603 transfer and 570 cm$^{-1}$ for either b601′-b608-b609 → a610-a611-a612 or b601′-b608-b609/b606-b607 → a613-a614 transfer.

While the model successfully accounts for many of the features in the 2DEV spectra of an electronically coupled system

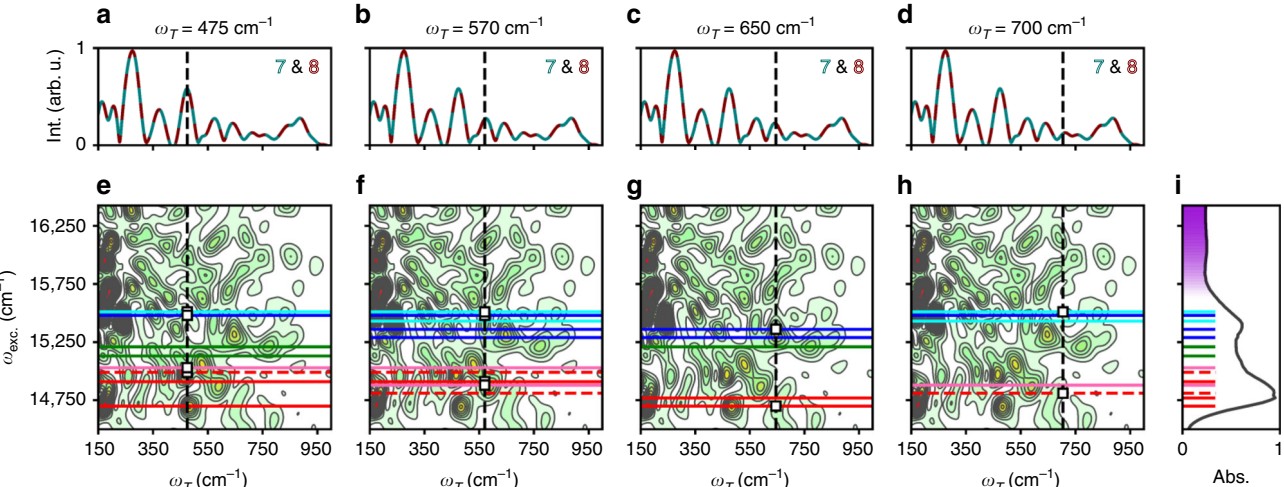

**Fig. 5 Assignments of observed beat frequencies. a–d** Magnitude of the cross-power spectrum between the CLS$_{exc.}$ of excited state absorption (ESA) seven and of ESA eight (peaks in the cross-power spectrum indicate shared frequencies). **e–h** Power spectrum along the excitation axis at a detection frequency of 1590 cm$^{-1}$ (through ESAs seven and eight, i.e., a Fourier transform of Fig. 6). Only peaks that survived the noise floor were plotted in (**e–h**) such that contour levels are drawn in 4% intervals starting from the top of the noise floor. The contour levels indicate peak intensity, where intensity is shown to increase as the colormap changes from green to red. Vertical dashed black lines indicate certain beat frequencies of interest: 475, 570, 650, and 700 cm$^{-1}$. Horizontal lines in (**e–h**) highlight excitonic states (colored according to Fig. 1) that fall along the beat frequencies of interest—the most important intersection points are accentuated by the white squares. **i** For clarity, the linear absorption spectrum of LHCII at 77 K is shown along with the excitonic states detailed in Fig. 1.

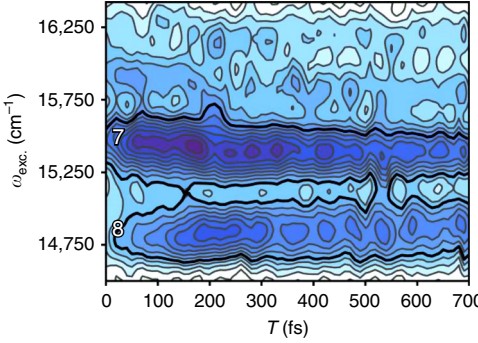

**Fig. 6 Spectral modulation along the excitation axis.** Spectral evolution along the excitation axis through excited state absorptions seven and eight, at a detection frequency of 1590 cm$^{-1}$, plotted as a function of waiting time. The colormap used here is identical to that in Fig. 2.

(including the emergence of ESA quartets, as well as peak amplitudes and CLSs that oscillate at the frequency of the excitonic energy gaps), it is still an enormous simplification of the actual spectrum of LHCII. The full understanding of electronic-vibrational dynamics is nontrivial and would require the inclusion of multiple vibrational DoF. This is clearly evident as demonstrated in Fig. 6, which shows the spectral evolution of ESAs seven and eight along the excitation axis (where a Fourier transform along the waiting time would create the intensity beat map introduced earlier in Fig. 5e–h). Upon inspection of Fig. 6, spectral modulation of ESAs seven and eight can be observed. The oscillatory features at the blue edge of eight are in phase with those at the red edge of seven and nearly completely out of phase with those at the blue edge of seven. Similar behavior is seen if slices are taken along the excitation axis through peaks five and six. The previous observation of this type of spectral modulation by Prall et al.[50] in a metal bridged phthalocyanine dimer was only accurately modeled by allowing for a nuclear modulation of the energies and interaction between participating electronic states, or in other words, by incorporating non-Condon effects, rather

than simpler wavepacket dynamics. The mechanism behind this type of behavior is driven by an opposite dependence of two electronic states on a common nuclear coordinate, as well as a nuclear dependence on the coupling between the two states. The observation of the spectral modulation effects depicted in Fig. 6 suggests a significant degree of mixing between the electronic and nuclear DoF in LHCII and inspection of Table 2 in ref. [51] shows that Chl *a* does indeed possess vibrations of very similar frequencies to those observed here and that all those observed have reasonable Huang-Rhys factors, making their likely involvement in mixing Chl *a* and *b* levels stronger.

**Concluding comments.** 2DEV spectroscopy provides a powerful way to untangle the connection between congested electronic spectra and complex ultrafast dynamics, such as EET. The connection of oscillatory features to specific excitation wavelengths allows their assignment to specific site states in LHCII, enabling a mapping of evolution in energy space to that in physical space. Our analysis reveals the critical role of the nuclear DoF in facilitating ultrafast energy transfer over moderately large energy gaps. In order to fully understand the role of non-Condon effects in photosynthetic light-harvesting, the manifestation of these effects in this newly developed technique will need to be developed, however, this work provides an important framework for elucidating the actual mechanistic role of vibronic interactions in these systems.

## Methods
**Sample preparation.** The isolation of thylakoid membranes was performed by using sucrose cushion[52] as described below. Deveined leaves were homogenized in 25 mM Tricine-KOH (pH 7.8), 400 mM NaCl, 2 mM MgCl$_2$, 0.2 mM benzamidine, and 1 mM ε-aminocaproic acid at 4 °C using a Waring blender for 30 s with max speed. The homogenate was filtrated through 4 layers of Miracloth, and the filtrate was centrifuged at 27,000 × *g* for 10 min at 4 °C. The pellet was resuspended in 25 mM Tricine-KOH (pH 7.8), 150 mM NaCl, 5 mM MgCl$_2$, 0.2 mM benzamidine, and 1 mM ε-aminocaproic acid. The suspension was loaded on sucrose cushion containing 1.3 M sucrose with 25 mM Tricine-KOH (pH 7.8), 15 mM NaCl, and 5 mM MgCl$_2$, which was overlaid on 1.8 M sucrose with 25 mM Tricine-KOH (pH 7.8), 15 mM NaCl, and 5 mM MgCl$_2$, and centrifuged at 131,500 × *g* for 30 min at 4 °C using a SW 32 Ti rotor (Beckman Coulter). Thylakoid membranes sedimented

in 1.3 M sucrose cushion were collected and washed with 25 mM Tricine-KOH (pH 7.8), 15 mM NaCl, and 5 mM MgCl$_2$, and centrifuged at $27,000 \times g$ for 15 min at 4 °C. The pellet was resuspended in 25 mM Tricine-KOH (pH 7.8), 0.4 M sucrose, 15 mM NaCl, and 5 mM MgCl$_2$, and centrifuged at $27,000 \times g$ for 10 min at 4 °C. The pellet was resuspended and used as purified thylakoid membranes.

The purified thylakoid membranes were resuspended in 25 mM HEPES-NaOH (pH 7.8) and centrifuged at $15,300 \times g$ for 10 min at 4 °C. The pellet was resuspended in 25 mM HEPES-NaOH (pH 7.8) at 2.0 mg Chl/mL and solubilized with 4% (w/v) n-dodecyl-α-D-maltoside (α-DM; Anatrace) for 30 min with gentle agitation on ice. The unsolubilized membranes were removed by centrifuging at $21,000 \times g$ for 5 min at 4 °C. The supernatant was filtrated through 0.22 μm filter using Durapore Ultrafree filters centrifuged at $10,000 \times g$ for 3 min at 4 °C. The 200 μL of filtered solubilized fraction was used for gel filtration chromatography using the ÄKTAmicro chromatography system with a Superdex 200 Increase 10/300 GL column (GE Healthcare) equilibrated with 25 mM HEPES-NaOH (pH 7.8) and 0.03% (w/v) α-DM at room temperature. The flow rate was 0.9 mL/min. The proteins were detected at 280 nm absorbance. The fraction separated from 10.0 to 10.3 mL contained trimeric LHCII proteins.

For the experiments, the maximum optical density of the LHCII sample in the visible was ~0.9 with a path length of 200 μm at 77 K (Optistate DN2, Oxford Instruments).

**Spectroscopic measurements**. Below we describe the 2DEV experimental setup[35] used in this work. A Ti:Sapphire oscillator and regenerative amplifier (Vitara-S, Legend Elite, Coherent) pumped a home-built visible NOPA and mid-IR OPA. The visible pump spectrum was centered at ~15,565 cm$^{-1}$ and spanned 14,350 ~ 16,775 cm$^{-1}$. As the experiments were performed in a partially collinear pump-probe geometry, a pulse shaper (Dazzler, Fastlite) was employed to generate the visible pump pulse pair and then control the relative initial time delay, $t_1$, (scanned from 0-100 fs in ~2.4 fs steps) and phase (the desired 2DEV signal was isolated with a $3 \times 1$ phase cycling scheme) between the pair[53,54]. The pulse shaper was also used to compress the visible pulses (~10 fs) and set the pump energy (~270 nJ). The visible pulses were focused into the sample to a spot size of 250 μm with a $f = 25$ cm silver coated 90° off-axis parabolic mirror. To remove the optical frequency of the pump, the data was collected in the fully rotated frame with respect to $t_1$. A Fourier transform was performed along $t_1$ to recover the excitation dimension of the spectra.

The mid-IR OPA produced an IR probe spectrum centered at ~1620 cm$^{-1}$. To account for shot-to-shot energy fluctuations in the IR, the probe beam was normalized by a reference. The IR probe and reference pulses had an energy of ~100 nJ and duration of ~50 fs. Both IR pulses were focused into the sample to a spot size of 200 μm with a $f = 15$ cm gold coated 90° off-axis parabolic mirror. After the sample, the IR probe and reference were dispersed with a spectrometer (Triax 180, Horiba) onto a dual-array 64 element HgCdTe detector (Infrared Systems Development).

Except for the NOPA, the entire setup was purged with dry air, free of CO$_2$ (Perkins Balston FT-IR Purge Gas Generator).

**Data processing**. To emphasize the lower frequency oscillations for presentation in the text, the data in Figs. 4–6 was subjected to a Savitzky-Golay filter[55] (essentially a ~1200 cm$^{-1}$ cutoff filter). An unfiltered version of Fig. 5 is provided in the Supplementary Note 3, which shows that filtering has no impact on the results discussed in this paper.

## Data availability

The data presented in this study are available from the corresponding author upon reasonable request.

## Code availability

The codes used for theoretical simulation in this work are available from the corresponding author upon reasonable request.

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

## Acknowledgements

This research was supported by the U.S. Department of Energy, Office of Science, Basic Energy Sciences, Chemical Sciences, Geosciences, and Biosciences Division. The authors thank Pallavi Bhattacharyya and Addison Schile for helpful discussions. E.A.A. acknowledges the support of the Berkeley Fellowship and the National Science Foundation Graduate Research Fellowship (Grant No. DGE 1752814). Y.Y. appreciates the support of the Japan Society for the Promotion of Science (JSPS) Postdoctoral Fellowship for Research Abroad.

## Author contributions

E.A.A. and G.R.F. conceived the research. E.A.A. and Y.Y. performed the 2DEV experiments. E.A.A. analyzed the experimental data and performed the theoretical calculations. E.A.A., Y.Y., and G.R.F. discussed the experimental and theoretical results. M.I. prepared the sample. E.A.A. and G.R.F. wrote the paper. E.A.A., Y.Y., M.I., K.K.N., and G.R.F. commented on the manuscript.

## Competing interests

The authors declare no competing interests.
