## [Peer Review File · Nature Communications]

[Editor's note: Some of the references to specific figure numbers in the Supplementary Information may be different in the final version.]

Reviewers' comments:

Reviewer #1 (Remarks to the Author):

The paper by Arsenault et al is very interesting. They have applied two-dimensional electronic-vibrational spectroscopy to reveal how selected vibrations (mainly chlorophyll vibrations) contribute to the coherent energy transfer between chlorophylls a and b. Especially, the dynamics of the CLSs very nicely reveals the values of the excitonic gaps (but this is working only for a large gaps that are not washed out by the disorder).

In the manuscript the authors state that 'el-vib mixing facilitates rapid b \rightarrow a conversion', but they explore the mixing in the 1500/cm region which is far from the b-a gap (600-700/cm). I agree that the dynamics of the 1500 vibrational peaks displays signatures of the exciton splittings, therefore vibronic dynamics at 1500/cm is a good marker for the exciton transfers (including identification of the exciton splittings), but the data do not show how the mixing with this vibration promotes b-a transfers. It would have been more interesting to probe at 600-700/cm.

In Fig 5e-h the points showing the exciton levels do not correspond to the peaks of the power spectra. So, the only way to assign the exciton transition is the correspondence between the CLS oscillating frequencies and the values of the splitting emerging from the exciton model of LHCII (I mean the power spectra in Fig 5e-h do not give any decisive information).

A detail, in fig. 3b E and E' are difficult to distinguish

About refs: the role of vibrations (including non-Condon el-vib mixing) in speeding up the EET/ET transfers was shown in PCCP 2017 for PSII-RC.

Reviewer #2 (Remarks to the Author):

This is a novel study using two-dimensional electronic vibrational (2DEV) spectroscopy to study energy transfer in the LHCII complex. The authors comment that the 2DEV experiment has the

potential to contribute to the debate about the possible importance of coherent energy transfer in photosynthesis. The data are high quality and the authors present a simplified model of the very complex LHClI system that reproduces the overall peak structure in the 2DEV spectra. However, the authors are making large claims in this paper that likely to be very controversial and that are not sufficiently substantiated. This is reminiscent of the author's 2007 Nature paper which assigned the modulations in two dimensional electronic spectroscopy (2DES) to electronic coherence. At that point 2DES was also a very new method and, as is 2DEV today, the interpretation of 2DES spectra was at an early stage. It is therefore imperative to proceed with caution in the interpretation of coherence in these new type of spectra. The 2007 work led to considerable confusion in the field that is still present today. There are important simplifications made in both the model and the interpretation of the 2DEV data that require further justification. Given the complexity of LHClI it would have been preferable to perform this study in a simpler system with fewer pigments.

i) The model presented in Figure 3 and the SI includes two coupled displaced oscillators with a single vibrational mode on the ground and excited state. The authors use the full Redfield quantum master equation with a Drude-Lorentz spectral density. The simulations show a 245 cm⁻¹ oscillation in the center line slope (CLS) that is out of phase for particular pairs of peaks. As in the 2007 Nature paper, the authors have neglected to consider how a pigment intramolecular vibration of 245 cm⁻¹ will modulate the CLS of different peaks, independently of any coherent energy transfer process. Raman and 2DES studies of chlorophyll a show an intramolecular mode very close to this frequency as well as other modulation frequencies observed by the authors (see for example Meneghin et al. Chemical Physics 2018, 514 132-140). The simulations must include the effect of adding an explicit 245 cm⁻¹ mode to evaluate its effect on the CLS and the cross-power spectrum and intensity beat maps in both the uncoupled and coupled monomer cases.

ii) The paper must clearly define coherent energy transfer. How does the CLS vary in the incoherent energy transfer case? Is an oscillation in the CLS a definitive measure of coherent energy transfer? How does it depend on electronic coupling strength?

iii) In the simulations, the peak pairs C'&C, and E'&E are shown to oscillate out of phase. However, in the LHClI data analysis, the authors have chosen to plot different peak pairs than the ones that show out of phase oscillations in the model. In Figure 4 they plot the CLS dynamics of peaks 5&7, 6&7 and 7&8. Each pair exhibits out of phase behavior, suggesting that pairs 5&6 should be in phase which is inconsistent with the model (which shows peaks E&E' out of phase).

iv) On page 10 the authors describe the creation of the intensity map shown in Figure 3g, stating that they take a slice at a detection frequency of 1545cm⁻¹ through peaks C and C'. The appropriate frequency appears to be closer to 1555cm⁻¹. How does the relative phase of various peak pairs change with detection frequency?

Reviewer #3 (Remarks to the Author):

The manuscript by Arsenault et al. represents the results of an application of two-dimensional electronic-vibrational spectroscopy to study ultrafast energy transfer in the pigments of higher plant LHCII complex. The title is very good and the work is novel and worth publishing. However, from the very beginning the text reads like a very heavy specialised presentation - not the style of a Nature-type journal. Moreover, whilst the authors stress the significance of LHCII complex and its abundance they do not explain why the measurements are performed at 77K - not ambient temperature. Further, the abstract lacks firm clarity. What was that debate about that it was mentioned in the abstract? Was it electronic coherence vs electron-vibrational interactions that enable femtosecond scale energy transfer? This has to be clearly stated. And in the conclusion: does actually this work show that there is no such a thing as electronic coherence in LHCII and all ultrafast energy transfer events can be described by the coupling of electron dynamics to nuclear motions? To me it is still not clear whether there is a proof that electron coherency takes place in nature - not under low temperature and laser excitation. After all, what is the use of such an artificial exercise described in the manuscript and the knowledge obtained? A potential for making artificial light harvesting devices? Certainly not for the understanding of light harvesting mechanisms and regulation in nature. These considerations that limit the breadth of this work have to be mentioned in this manuscript.

Reviewer #4 (Remarks to the Author):

The authors use 2D EV spectroscopy to map excitation energy transfer in LHCII at 77K. The results of this paper add to the literature describing the role of vibronic coupling in ultrafast energy transfer of natural light harvesting systems. In this paper, higher lying vibronic states and non-Condon effects are implicated in the rapid electron transfer.

The authors should address the following points in a revised manuscript before publication:

- 1) It would be useful to see the FTIR spectrum of the LHC II complex in the 1500 – 1750 cm^{-1} region. What is the assignment of the vibrational frequencies that are observed in the 2D EV spectrum? They are simply labeled as belonging to Chl a or Chl b in the text. A better description is necessary. What are the ground state frequencies associated with the ESAs observed?
- 2) A discussion regarding the fact that only two excitonic bands are seen in the 2D EV spectra is warranted. Fig 1 b shows 14 excitonic states. What is the reason that only two main excitonic peaks are observed in the 2D EV spectrum? In ref. 9, the 2D ES experiment was able to identify ~ 14 excitonic states using the beat frequencies.
- 3) What are the contour levels in Fig. 2? Where is the noise floor? A color bar would be useful.
- 4) The first line on Page 8 refers to the Qy states. What is this referring to?

- 5) What are the higher lying vibronic states? Have they been studied earlier with transient absorption, transient IR or 2D ES experiments?
- 6) The authors use CLS to understand energy transfer between the excitonic manifolds. An example of the CLS should be provided in the supplementary section so the readers can ascertain the quality of the fit for the slope. Across what w_1 pixels is the CLS extracted? How does the standard error of the fit change for different data runs?
- 7) Why is CLS a better way to characterize energy transfer instead of directly using the amplitudes of the cross-peaks as a function of T as shown in Fig 1 e?
- 8) What is the noise floor on the beat frequency maps shown in Figures 5e)-f)?
- 9) Are different beat frequencies observed at different detection frequencies?
- 10) Why is the discussion focused on 475, 570, 650 and 700 cm^{-1} ? What about the $\sim 250 \text{ cm}^{-1}$ peak seen in Figures 5 a-d?
- 11) Figure 6 shows the spectral evolution along the excitation axis for different detection frequencies and oscillations are discussed. For the readers to believe these oscillations, a Fourier transform should be performed and displayed in the same Figure.
- 12) The data shown in Figure 6 is given as evidence of non-Condon effects by referencing an earlier study by the authors (Prall et al, Ref. 44). That particular study was based on transient absorption and did not include an IR probe. Would the same effect be seen in a 2D EV experiment?
- 13) Non-Condon effects in 2D EV experiments will be non-trivial and could be seen in amplitude modulation and spectral modulation of GSB and ESA peaks. Coupling with low frequency modes will also complicate the observations. Before the authors claim Figure 6 as evidence of Non-Condon effects, they should simulate it with a simple model.

Response to Reviewers

We thank the reviewers for their careful reading of our paper and we appreciate that they all found our paper interesting and worthwhile. Their comments and questions sent us back to the laboratory to demonstrate experimentally that vibrational wavepackets in modes not anharmonically coupled to the detected modes do not influence the 2DEV spectra.

Many of their comments suggest that we need to improve the clarity with which we present the 2DEV experiment (on page 4), as well our conclusions. In response, we have made significant changes to the manuscript. These include the addition of a control 2DEV experiment (now provided in the SI), in order to clarify the role of vibrational coherences in this relatively new experimental technique, and the careful determination of our noise floor (Reviewer #4), such that we only show modulations above this limit. Changes to all of the figures have also been made to improve clarity. An expanded discussion concerning the 2DEV experiment itself has been added to the manuscript, as well as a new discussion concerning the motivation for choosing the particular detection frequency range used in this work. We also recognize that, as was pointed out by several reviewers, it is important to proceed with caution when commenting on the role and manifestation of non-Condon effects in the 2DEV spectra of electronically coupled systems. In response to this, we have chosen to conclude the manuscript in a less assertive manner regarding the role of these effects and instead comment on the significant potential that 2DEV has in terms providing actual mechanistic insight into the role of quantum beats in photosynthetic light-harvesting. Below we address each of the reviewers' comments point-by-point. Changes to the manuscript and Supplemental information are indicated in red.

Reviewer #1:

The paper by Arsenault et al is very interesting. They have applied two-dimensional electronic-vibrational spectroscopy to reveal how selected vibrations (mainly chlorophyll vibrations) contribute to the coherent energy transfer between chlorophylls a and b. Especially, the dynamics of the CLSs very nicely reveals the values of the excitonic gaps (but this is working only for a large gaps that are not washed out by the disorder).

In the manuscript the authors state that 'el-vib mixing facilitates rapid ba conversion', but they explore the mixing in the 1500/cm region which is far from the b-a gap (600-700/cm). I agree that the dynamics of the 1500 vibrational peaks displays signatures of the exciton splittings, therefore vibronic dynamics at 1500/cm is a good marker for the exciton transfers (including identification of the exciton splittings), but the data do not show how the mixing

with this vibration promotes b-a transfers. It would have been more interesting to probe at 600-700/cm.

The modes in the 1500-1700 cm^{-1} region do not promote energy transfer, rather these modes are spectators that report on the energy transfer dynamics. These modes were chosen as probes because they are highly local and thus maintain to a high degree their site character. One major advantage of probing in this region, as opposed to 600-700 cm^{-1} , is that when the system exhibits little to no anharmonic coupling between low and high frequency vibrational modes, the dynamics observed via probing the higher frequency modes are free from the modulation effects caused the creation of low(er) frequency wavepackets. By probing in the lower frequency region, these effects would cause additional complication and would have to be considered carefully. A more detailed discussion has been added to the SI (pgs. S4-S7). Additionally, we have clarified this in the manuscript on pg. 5 and the top of pg. 6.

In Fig 5e-h the points showing the exciton levels do not correspond to the peaks of the power spectra. So, the only way to assign the exciton transition is the correspondence between the CLS oscillating frequencies and the values of the splitting emerging from the exciton model of LHCII (I mean the power spectra in Fig 5e-h do not give any decisive information).

The power spectra are very congested and thus we provided a method (using the CLS frequencies and intensity beat frequencies together) by which to unravel the various energy transfer pathways simultaneously contributing to the spectra. Without the power spectra in Fig 5e-h, we would not be able to determine which excitonic levels were contributing. To achieve this; however, we used the experimentally determined excitonic energy levels from Ref. 9 rather than those from the model in Ref. 4.

A detail, in fig. 3b E and E' are difficult to distinguish

We have enlarged the size of the label and changed its position, as well as the colormap in Fig. 3b to make the prime more distinguishable.

About refs: the role of vibrations (including non-Condon el-vib mixing) in speeding up the EET/ET transfers was shown in PCCP 2017 for PSII-RC.

We thank the reviewer for the reference, which we believe to be: V. I. Novoderezhkin, E. Romero, J. Prior, and R. van Grondelle *Phys. Chem. Chem. Phys.* **19**, 5195-5208 (2017). This paper, which demonstrates that the mixing of vibrations resonant with exciton-charge transfer (CT) or CT-CT energy gaps in the PSII reaction center can give rise to efficient charge separation, was added as Ref. 34 to the manuscript.

Reviewer #2:

This is a novel study using two-dimensional electronic vibrational (2DEV) spectroscopy to study energy transfer in the LHCII complex. The authors comment that the 2DEV experiment has the potential to contribute to the debate about the possible importance of

coherent energy transfer in photosynthesis. The data are high quality and the authors present a simplified model of the very complex LHCI system that reproduces the overall peak structure in the 2DEV spectra. However, the authors are making large claims in this paper that likely to be very controversial and that are not sufficiently substantiated. This is reminiscent of the author's 2007 Nature paper which assigned the modulations in two dimensional electronic spectroscopy (2DES) to electronic coherence. At that point 2DES was also a very new method and, as is 2DEV today, the interpretation of 2DES spectra was at an early stage. It is therefore imperative to proceed with caution in the interpretation of coherence in these new type of spectra. The 2007 work led to considerable confusion in the field that is still present today. There are important simplifications made in both the model and the interpretation of the 2DEV data that require further justification. Given the complexity of LHCI it would have been preferable to perform this study in a simpler system with fewer pigments.

We appreciate that 2DEV and its analysis are not fully mature. The reviewer's comments regarding interpretation caused us to carry out further experiments to demonstrate, as commented above, that strong wavepacket beating in a 2DES measurement does not simply carry over to a 2DEV measurement. The new data in the SI show the strong beats exhibited by cresyl violet (Refs. 12-14 in SI) do not appear in the 2DEV spectrum of this molecule when the electronic excitation was identical to that used in a previous 2DES studies (Ref. 12 in the SI).

i) The model presented in Figure 3 and the SI includes two coupled displaced oscillators with a single vibrational mode on the ground and excited state. The authors use the full Redfield quantum master equation with a Drude-Lorentz spectral density. The simulations show a 245 cm⁻¹ oscillation in the center line slope (CLS) that is out of phase for particular pairs of peaks. As in the 2007 Nature paper, the authors have neglected to consider how a pigment intramolecular vibration of 245 cm⁻¹ will modulate the CLS of different peaks, independently of any coherent energy transfer process. Raman and 2DES studies of chlorophyll a show an intramolecular mode very close to this frequency as well as other modulation frequencies observed by the authors (see for example Meneghin et al. Chemical Physics 2018, 514 132-140). The simulations must include the effect of adding an explicit 245 cm⁻¹ mode to evaluate its effect on the CLS and the cross-power spectrum and intensity beat maps in both the uncoupled and coupled monomer cases.

We agree that full analysis of our spectra requires the inclusion of multiple vibrational degrees of freedom. However, we feel that at this stage the understanding of this new spectroscopy is best facilitated by the simplest model capable of representing much of the observed spectral features and dynamics. Our model demonstrates the natural appearance of "quartets" in the spectra and shows that the peak amplitudes and CLS will oscillate at the frequencies of excitonic energy gaps, regardless of the inclusion of additional vibrational modes. As noted above, the new control experiments in the SI demonstrate that simple wavepacket motion in uncoupled modes does not appear in the 2DEV spectra focused in the mid-IR. Nonetheless, the full understanding of electronic-vibrational dynamics is obviously complicated and we have de-emphasized the discussion of non-Condon effects. These changes are reflected in the second half of the abstract, the

concluding sentence of the Introduction (pg. 4), additional text on pg. 14 (first two sentences of the first paragraph), and the last sentence of the Concluding Comments (pg. 15).

ii) The paper must clearly define coherent energy transfer. How does the CLS vary in the incoherent energy transfer case? Is an oscillation in the CLS a definitive measure of coherent energy transfer? How does it depend on electronic coupling strength?

Thank you for noting this potential point of confusion. In the revised manuscript we have more carefully considered our word choice. We now simply refer to energy transfer without a modifier, leaving the discussion of “coherent” energy transfer to future work.

iii) In the simulations, the peak pairs C’&C, and E’&E are shown to oscillate out of phase. However, in the LHCII data analysis, the authors have chosen to plot different peak pairs than the ones that show out of phase oscillations in the model. In Figure 4 they plot the CLS dynamics of peaks 5&7, 6&7 and 7&8. Each pair exhibits out of phase behavior, suggesting that pairs 5&6 should be in phase which is inconsistent with the model (which shows peaks E&E’ out of phase).

In order to ease the technical aspect of the analysis, we have decided to omit the discussion on phase relationships in the CLS dynamics, especially as the discussion is centered almost completely around shared frequencies. These changes are reflected on pg. 10, as well as in Fig. 5. Of course, LHCII is more complicated than the model system and so multiple energy transfer steps can occur simultaneously, thus translating to more complicated phase relationships (with many oscillatory frequencies present in the CLS), which leads to combinations of in and out of phase beats between multiple pairs of excitons.

To address the reviewer’s specific question, we have included additional pairs of time domain CLS dynamics in Fig. 4 to match those shown in Fig. 3. They highlight the presence of many different oscillatory frequencies and show that, for reasons just discussed, none of these pairs exhibit fully in or fully out of phase behavior.

iv) On page 10 the authors describe the creation of the intensity map shown in Figure 3g, stating that they take a slice at a detection frequency of 1545cm⁻¹ through peaks C and C’. The appropriate frequency appears to be closer to 1555cm⁻¹.

We thank the reviewer for a careful reading of the manuscript. The manuscript now contains the correct frequency, 1555 cm⁻¹ (pg. 11).

How does the relative phase of various peak pairs change with detection frequency?

As stated in our response to iii) above, we have decided to omit the discussion on phase relationships in CLS dynamics. However, we have included additional pairs of time domain CLS dynamics in Fig. 4.

Reviewer #3:

The manuscript by Arsenault et al. represents the results of an application of two-dimensional electronic-vibrational spectroscopy to study ultrafast energy transfer in the pigments of higher plant LHCII complex. The title is very good and the work is novel and worth publishing. However, from the very beginning the text reads like a very heavy specialised presentation - not the style of a Nature-type journal.

Moreover, whilst the authors stress the significance of LHCII complex and its abundance they do not explain why the measurements are performed at 77K - not ambient temperature.

The reason that these experiments were performed at 77 K was largely practical—at these temperatures, sample integrity can be maintained for the duration of these lengthy experiments. This choice was made because it is well-documented in the literature that energy transfer rates are independent of temperature (this is summarized for a similar photosynthetic antenna complex to LHCII in the following reference: van der Laan, et al. *Chem. Phys. Lett.* **170**, 231-238 (1990)).

Further, the abstract lacks firm clarity. What was that debate about that it was mentioned in the abstract? Was it electronic coherence vs electron-vibrational interactions that enable femtosecond scale energy transfer? This has to be clearly stated. And in the conclusion: does actually this work show that there is no such a thing as electronic coherence in LHCII and all ultrafast energy transfer events can be described by the coupling of electron dynamics to nuclear motions? To me it is still not clear whether there is a proof that electron coherency takes place in nature - not under low temperature and laser excitation. After all, what is the use of such an artificial exercise described in the manuscript and the knowledge obtained? A potential for making artificial light harvesting devices? Certainly not for the understanding of light harvesting mechanisms and regulation in nature. These considerations that limit the breadth of this work have to be mentioned in this manuscript.

We have shortened the abstract to meet the journal guidelines and moved the discussion of the debate to the introduction (bottom of pg. 2). Additionally, we have reworked the Introduction to stress the potential of 2DEV for providing a connection between spectroscopic observations and actual mechanistic function (bottom of pg. 3 and middle of pg. 4).

Reviewer #4:

The authors use 2D EV spectroscopy to map excitation energy transfer in LHCII at 77K. The results of this paper add to the literature describing the role of vibronic coupling in ultrafast energy transfer of natural light harvesting systems. In this paper, higher lying vibronic states and non-Condon effects are implicated in the rapid electron transfer. The authors should address the following points in a revised manuscript before publication:

1) It would be useful to see the FTIR spectrum of the LHC II complex in the 1500 – 1750 cm⁻¹ region.

The FTIR spectrum of LHCII in the 1500-1700 cm⁻¹ region is broad and essentially structureless, as seen in: W. I. Gruszecki, et al. *BBA-Energetics* **1757**, 1504-1511 (2006). The reason that we see discrete features is because the spectra result from population differences, not the absolute population.

What is the assignment of the vibrational frequencies that are observed in the 2D EV spectrum? They are simply labeled as belonging to Chl a or Chl b in the text. A better description is necessary. What are the ground state frequencies associated with the ESAs observed?

Many of the features not discussed in the text have been assigned previously and discussed in detail in Ref. 7. We have added a note about this in the text at the top of pg. 7. Through help from the model, we have for the first time shown a more detailed origin of the two ESA quartets in the 2DEV spectra. The ground state frequencies associated with these eight ESAs are unknown, due to the fact that the nature of these ESAs has previously been completely unknown. The complicated electronic structure arising from the many chlorophyll pigments in the LHCII trimer makes it very difficult to assign these ESAs beyond stating the amount of Chl *a* versus Chl *b* character that they have. As a result of this, those with mainly Chl *a* or Chl *b* character are assigned as “Chl *a*” or “Chl *b*” accordingly.

2) A discussion regarding the fact that only two excitonic bands are seen in the 2D EV spectra is warranted. Fig 1 b shows 14 excitonic states. What is the reason that only two main excitonic peaks are observed in the 2D EV spectrum? In ref. 9, the 2D ES experiment was able to identify ~14 excitonic states using the beat frequencies.

As shown in the absorption spectra now placed at the top of Fig. 2, there are many closely lying excitonic levels in LCHII, resulting from the large number of electronically coupled Chls. Even at 77 K, this results in the appearance of two excitonic bands, one with mainly Chl *a* character and one with mainly Chl *b* character. The 14 excitonic states can; however, be identified through a Fourier transform, as elucidated in Fig. 5e-h and as done in Ref. 9. The clear separation of the Chl *a* and Chl *b* regions along the excitation axis (as compared to the electronic absorption spectrum) arises from the detection method, i.e. the observation via vibrational transitions. This also clearly emerges in the model spectra in Fig. 3.

3) What are the contour levels in Fig. 2? Where is the noise floor? A color bar would be useful.

Fig. 2 has been updated to include the absorption spectrum of LHCII, together with the excitonic energy levels and a color bar. The colormap has also been updated such that the noise floor falls within ± 1 contour level away from zero, making the GSB and ESA features versus the noise floor more apparent. Additionally, the figure caption has been updated to include the contour level spacing.

4) The first line on Page 8 refers to the Q_y states. What is this referring to?

Q_y is the standard nomenclature for the S₁ state of Chl molecules. We prefer to stay with this usage and have added clarifying text to the caption of Fig. 1 (we excite the Q_y bands of LHCII for this experiment).

5) What are the higher lying vibronic states? Have they been studied earlier with transient absorption, transient IR or 2D ES experiments?

To the best of our knowledge these have not received significant attention and our manuscript is the first attempt at assigning them further and discerning their role in the energy transfer process. This is the first substantial discussion of these vibronic bands because they are more pronounced in our 2DEV spectra than in other experiments, as a result of the unique combination of electronic excitation and visible detection. As stated in the manuscript, analysis of the 2DEV spectra made it possible to discern that they are composed of mixed Chl *a* and Chl *b* character. Even more important than the assignment; however, is the observation that they participate in rapid energy transfer to the lower lying Chl *a* and Chl *b* states.

6) The authors use CLS to understand energy transfer between the excitonic manifolds. An example of the CLS should be provided in the supplementary section so the readers can ascertain the quality of the fit for the slope. Across what w1 pixels is the CLS extracted? How does the standard error of the fit change for different data runs?

We provided the standard error in Fig. 4, but we apologize that it was difficult to see. In response, we have updated Fig. 4 for clarity.

The CLS is extracted across the excitation axis in accordance with the width of the excitonic band for a given feature (Refs. 36 and 37 in the manuscript). For example, the width of feature 8 along the excitation axis would be equivalent to the width of that mainly Chl *a* excitonic band.

The standard error of the fit is dependent upon the quality of the 2DEV data. The standard errors presented in Fig. 4 are as to be expected for a good data set. Another example of the standard error of CLS data can now be seen in the additional experimental data for the control experiment presented in the SI.

7) Why is CLS a better way to characterize energy transfer instead of directly using the amplitudes of the cross-peaks as a function of T as shown in Fig 1 e?

The CLS has one major advantage in that it is particularly sensitive to energy transfer, such that it is a sort of “exciton coherence-specific” measurement because it is free of population dynamics which greatly complicate the cross-peak amplitudes. However, we would like to stress that the best method of characterization is through a combination of both the CLS *and* peak amplitude dynamics if the goal is to both determine beat frequencies *and* assign them to the interplay between specific energy levels. We have added a note on this in the manuscript at the bottom of pg. 10.

8) What is the noise floor on the beat frequency maps shown in Figures 5e)-f)?

Fig. 5e-h has been amended such that only peaks that survive the noise floor are plotted. A note of this has also been added to the figure caption. Additionally, we have added a figure to the SI, Fig. S9, which more explicitly compares the average noise floor to the beat frequencies. The noise floor was determined by calculating the average power spectrum of the experimental noise.

9) Are different beat frequencies observed at different detection frequencies?

Additional intensity profiles belonging to features 5-8 have been provided in Fig. 2. For clarity and because of the sheer amount of data, the main discussion in the text is still limited to peaks 7 and 8.

10) Why is the discussion focused on 475, 570, 650 and 700 cm⁻¹? What about the ~250 cm⁻¹ peak seen in Figures 5 a-d?

The lower frequency region has not been explicitly analyzed because this region is too congested. More specifically, the concurrent loss of frequency resolution at lower frequencies and large number of closely spaced excitonic states makes it very difficult to explicitly assign the beats at lower frequencies. Comments on this have been added to pg. 12 of the manuscript.

11) Figure 6 shows the spectral evolution along the excitation axis for different detection frequencies and oscillations are discussed. For the readers to believe these oscillations, a Fourier transform should be performed and displayed in the same Figure.

Fig. 6 has been updated so that it now shows spectral evolution along the detection axis through features 7 and 8, rather than 5 and 6, so that the Fourier transform of Fig. 6 directly yields the power spectra shown in Fig 5e-h. We have also explicitly stated this on pg. 14 of the manuscript.

12) The data shown in Figure 6 is given as evidence of non-Condon effects by referencing an earlier study by the authors (Prall et al., Ref. 44). That particular study was based on transient absorption and did not include an IR probe. Would the same effect be seen in a 2D EV experiment?

13) Non-Condon effects in 2D EV experiments will be non-trivial and could be seen in amplitude modulation and spectral modulation of GSB and ESA peaks. Coupling with low frequency modes will also complicate the observations. Before the authors claim Figure 6 as evidence of Non-Condon effects, they should simulate it with a simple model.

12) and 13): As similar concerns were shared by other reviewers, we have more carefully considered our conclusion regarding the role of non-Condon effects, as stated at the beginning of our response and in the last sentence of our response to comment i) of Reviewer #2.

REVIEWERS' COMMENTS:

Reviewer #2 (Remarks to the Author):

The authors have made significant changes to the manuscript that address most of the concerns raised by the reviewers and their claims are now more convincingly supported by their data. Several concerns remain that should be addressed prior to publication.

1) The addition of the control study on cresyl violet is helpful for demonstrating that the observed oscillations in the center line slope (CLS) are not a result of vibrational wavepacket dynamics. However, the choice of cresyl violet seems odd given that the authors have previously published data on chlorophylls a and b which are the more relevant controls here. Similar to cresyl violet, they have also been shown to exhibit vibrational coherences. What is the reasoning for choosing cresyl violet? The authors should include the CLS analysis of their chlorophyll data or provide clear justification for choosing cresyl violet over chlorophyll as a control.

2) As pointed out by the authors, the insensitivity of 2DEV to vibrational wavepacket dynamics makes it appealing. Another useful related approach is polarization-dependent 2D electronic spectroscopy as demonstrated by Westenhoff et al (JACS, 2012, 134, 16484). The authors should cite this work.

Reviewer #4 (Remarks to the Author):

The authors have carefully considered the reviewer comments and have made significant improvements to the manuscript. I recommend publication in the current form.

Response to Reviewers

Reviewer #2:

The authors have made significant changes to the manuscript that address most of the concerns raised by the reviewers and their claims are now more convincingly supported by their data. Several concerns remain that should be addressed prior to publication.

1) The addition of the control study on cresyl violet is helpful for demonstrating that the observed oscillations in the center line slope (CLS) are not a result of vibrational wavepacket dynamics. However, the choice of cresyl violet seems odd given that the authors have previously published data on chlorophylls a and b which are the more relevant controls here. Similar to cresyl violet, they have also been shown to exhibit vibrational coherences. What is the reasoning for choosing cresyl violet? The authors should include the CLS analysis of their chlorophyll data or provide clear justification for choosing cresyl violet over chlorophyll as a control.

We chose cresyl violet specifically because it is well known to exhibit strong vibrational coherences at well-defined frequencies. The combination of strong signals and no spectral congestion makes cresyl violet an ideal control system—the presence of such signals would be easily interpretable or the lack thereof would be undeniable. To clarify this, we amended the second to last sentence on the bottom of page 6. We feel that this is a more convincing demonstration than the absence of beats in chlorophyll where we already know they are weak in visible pump-visible probe spectra.

2) As pointed out by the authors, the insensitivity of 2DEV to vibrational wavepacket dynamics makes it appealing. Another useful related approach is polarization-dependent 2D electronic spectroscopy as demonstrated by Westenhoff et al (JACS, 2012, 134, 16484). The authors should cite this work.

We thank the reviewer for the reference to this work. It has been added as reference 45 to the manuscript. Additionally, a note that refers specifically to this reference has been added at the bottom of page 3.